# Experimental demonstration of high space compression by optical spaceplates

Ryan Hogan [1,2] ✉, Yaryna Mamchur[2], R. Margoth Córdova-Castro[2], Graham Carlow[3], Brian T. Sullivan[3], Orad Reshef [2], Robert W. Boyd [2,4] & Jeff S. Lundeen [2]

The large physical size of optical imaging systems is one of the greatest constraints on their use, limiting the performance and deployment of a range of systems from telescopes to mobile phone cameras. Spaceplates are nonlocal optical devices that compress free-space propagation into a shorter distance, paving the way for more compact optical systems, potentially even thin flat cameras. Here, we demonstrate an engineered optical spaceplate and experimentally observe the highest space compression ratios yet demonstrated in any wavelength region, up to $\mathcal{R} = 176 \pm 14$, which is 29 times higher than any previous device. Our spaceplate is a multilayer stack, a well-established commercial fabrication technology that supports mass production. The versatility of these stacks allows customization of the spaceplate's bandwidth and angular range, impossible with previous optical experimental spaceplates made of bulk materials. With the appropriate choice of these two parameters, multilayer spaceplates have near-term applications in light detection and ranging (LIDAR) technologies, retinal scanners, endoscopes, and other size-constrained optical devices.

As the demand for more compact optical imaging systems grows, new approaches to shrinking their physical footprint have become essential. Until recently, research has focused mainly on reducing the thickness of optical elements, with notable progress in metasurfaces[1], diffractive lenses[2,3], and refractive Fresnel lenses[4]. However, shrinking these elements[5,6] only addresses one part of the challenge of optical miniaturization: the propagation of light across the physical distance between elements is equally important for image formation. These distances must also be reduced to compactify the entirety of an optical system, such as a telescope or microscope. Ref. 7,8 proposed a new approach to reducing the total length of optical systems, the spaceplate: a nonlocal optical device that compresses the propagation of light.

As shown in Fig. 1a, a spaceplate can be understood by its action on incoming rays of light. A ray incident at angle $\theta$ will propagate through an air path of length $d_{\text{eff}}$ (i.e., a slab of free space), exit at the same angle $\theta$, and be displaced transversely by distance $w = d_{\text{eff}} \tan \theta$ by simple geometry. The spaceplate provides precisely this transformation but in a shorter length $d$ and, in this sense, compresses the needed space by the ratio $\mathcal{R} = d_{\text{eff}}/d$[7].

Consider the effect of the spaceplate on the rays focused by a lens, as shown in Fig. 1b. The transverse ray displacement $w$ removes the need for $d_{\text{eff}}$ of propagation length after the lens. With the spaceplate thickness $d$ still present, the distance between the lens and the focus is reduced by $\Delta = d_{\text{eff}} - d$. Typically, one would shorten the distance to the focus by simply reducing the lens's focal length; however, doing this would be accompanied by a change in the ray angles that changes image magnification, which may not be desired. In contrast, adding a space plate to a camera will not change its magnification. In this way, the spaceplate breaks the link between magnification and focal length and thus relaxes the limitations on all dimensions of an imaging system. For example, combining the

---

[1]National Key Laboratory of Solid State Microstructures, School of Physics, and Collaborative Innovation Center of Advanced Microstructures, Nanjing University, Jiangsu, China. [2]Nexus for Quantum Technologies, Department of Physics, University of Ottawa, Ottawa, ON, Canada. [3]Iridian Spectral Technologies, Ottawa, ON, Canada. [4]Institute of Optics, University of Rochester, Rochester, NY, USA. ✉e-mail: ryan.hogan@nju.edu.cn

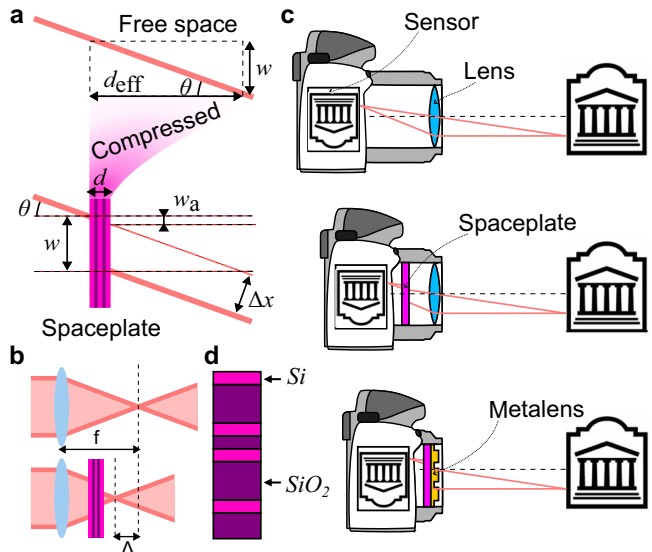

**Fig. 1 | The action of a spaceplate. a** An effective length of free space, $d_{eff}$, is replaced by a spaceplate of a thickness, $d$. In both cases, for a light incident at an angle $\theta$, the beam is shifted by $w$, resulting in a lateral shift $\Delta x$. **b** When the spaceplate is placed in front of a lens, the focal plane experiences a shift $\Delta = d_{eff} - d$ towards the device. **c** With the use of both a metalens and a spaceplate, flat and thin optical systems are possible. **d** A unit cell of multilayer spaceplate sample FPC2, alternating between hydrogenated amorphous silicon (Si) and silica ($SiO_2$).

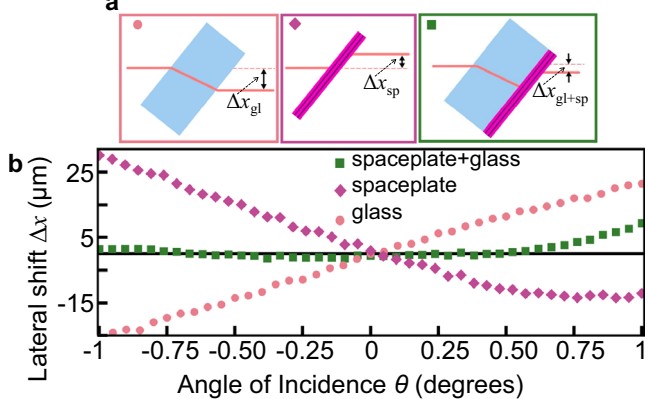

**Fig. 2 | Demonstration of a thin spaceplate cancelling the lateral beam shift created by a tilted glass plate. a** A positive lateral shift due to refraction in a glass plate (left) versus a negative shift from a freestanding spaceplate (middle), and our measurement scenario: a multilayer spaceplate deposited on glass (right). **b** Experimentally measured lateral shift for device FPC2 at the wavelength of $\lambda = 1532.905$ nm for scenarios: **a**-left, $\Delta x_{gl}$ (pink circles); **a**-right, $\Delta x_{gl+sp}$ (green squares); and **a**-middle, the difference in their shifts, $\Delta x_{gl+sp} - \Delta x_{gl} = \Delta x_{sp}$ (purple diamonds). i.e., the lateral shift from the spaceplate alone. Notably, the green curve indicates almost no beam movement as the angle is changed, implying that the spaceplate is fully negating the lateral shift of the 260 times thicker glass.

spaceplate with a flat lens opens the path to the creation of a thin flat monolithic camera (see Fig. 1c)[7].

Since their conceptual introduction and first experimental demonstration, spaceplates have attracted significant interest and innovation. Proposed theoretical design methodologies for engineered spaceplates include inverse design of multilayer stacks[7,9], coupled resonators[10], and photonic crystals[8,11]. Theoretical limits on the performance of spaceplates have also been studied[12,13]; Ref. 12 showed that a spaceplate that operates throughout the visible spectrum and is made from physically realizable materials might achieve an $\mathcal{R}$ as high as 8 up to a maximum angle of incidence $\theta_{max}$ of 90°[12]. Two types of spaceplates have been experimentally demonstrated in the visible and near-infrared wavelength region, both composed of bulk crystals or optics; the first demonstration of the spaceplate concept only replaced space in an $n > 1$ medium[7], while the second replaced on the order of a meter of vacuum propagation but only up to a small $\theta_{max}$ ($\mathcal{R} < 15.6$ and $\theta_{max} < 3.4°$)[14]. An engineered high-compression-ratio spaceplate has been experimentally demonstrated solely in the microwave region (using a multilayer design with $\mathcal{R} < 6$ and $\theta_{max} = 19.5°$)[15]. For imaging purposes or use in integrated photonics, a spaceplate operating in the optical wavelength region would be desirable.

This work presents a multilayer spaceplate engineered to achieve high compression ratios in the optical wavelength region. As a first example that demonstrates its unusual action of reducing the propagation length, consider a light beam transmitted through a glass plate. The beam will move from side to side as the glass is tilted back and forth. On the other hand, as Fig. 2a shows, a spaceplate will move a beam in the opposite direction to that expected of the glass or any ordinary material. In Fig. 2b, we present such experimental observations on a spaceplate with $\mathcal{R} = 96$. In particular, we show that this $\mathcal{R}$ is so large that even though the spaceplate is only 11.51 microns thick, when placed on top of a 3-mm-thick glass plate, it cancels the beam movement despite being 260 times thinner, albeit over a limited range of angles. We present other similarly striking observations in more detail in the Results section.

We experimentally demonstrate two types of multilayer designs near a wavelength of 1550 nm made from alternating layers of silica ($SiO_2$, $n = 1.44$) and amorphous silicon (deposited with hydrogen, a-Si:H, $n = 3.2$), which are common materials that are compatible with volume manufacturing in commercial fabrication facilities for multilayer stacks. We design the multilayer to create the nonlocal phase response associated with the Fourier transfer-function of free space[16]. A closely related nonlocal response, namely an angle-dependent phase shift, has long been shown to cause a transverse beam shift $w$[17], with some optical multilayer devices exhibiting large shifts[18,19]. However, a spaceplate is defined not merely by its ability to cause a beam shift but by its capacity to produce this shift with the angular dependence required to replace free space, $w = d_{eff} \tan \theta$ (as shown in Fig. 1a). For more information, see Supplementary Method 8 and Supplementary Fig. 6.

Our two types of multilayer designs use different algorithms for choosing the layer thickness to achieve the spaceplate response. The first design (GD) is an inversely-designed aperiodic structure found from gradient-descent optimization as detailed in ref. 9. The second design (FPC) is a periodic structure consisting of a series of identical Fabry-Pérot cavities separated by $\lambda/2$ layers, motivated by ref. 10. Note that, the FPC design is distinct from the standard Fabry-Pérot filter design, which uses $\lambda/4$ coupling layers between the cavities (for further details see Supplementary Method 1, Supplementary Fig. 1, Supplementary Tables 1 and 2). Due to the versatility of the multilayer platform, designs can be tuned to achieve a specific angular range, bandwidth, or compression ratio, although there are trade-offs between these performance parameters[12]. This design versatility makes multilayer spaceplates suitable for a wide range of applications, including LIDAR[20] and advanced imaging techniques[21,22].

## Results

### Observation of the focal shift

The most compelling application of spaceplates is for miniaturizing imaging systems. In this section, we observe the shortening of image and focal distances due to the presence of a spaceplate. All four of our multilayer structures, which range from 2.48 to 13.42 microns thick, are deposited on fused-silica glass substrates that are nominally 3 mm thick (exact numbers are given in Supplementary Table 1). The

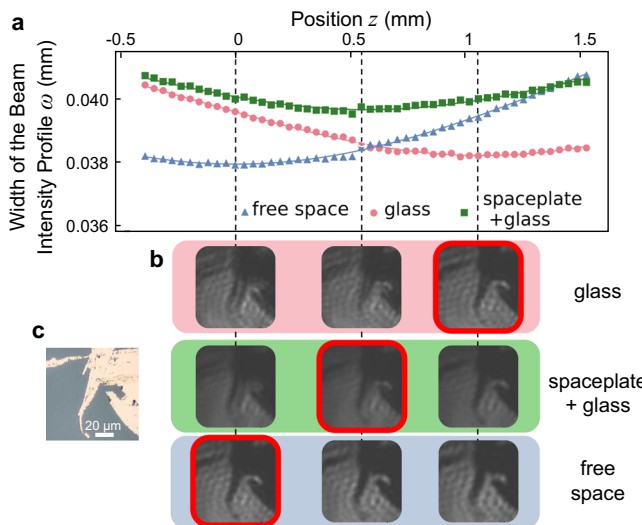

**Fig. 3 | Shifts of the focal plane and imaging. a** The width $\omega$ of a focusing beam with wavelength $\lambda = 1546.85$ nm, as in Fig. 1**b**, at different positions $z$ along the beam propagation direction for free space (blue triangles), a spaceplate (device FPC2) on glass (green squares), and only glass (pink circles), along with a corresponding fitted curve from standard Gaussian beam theory (Eq. 3.1-8[16]). The minimum width is at the focus position (dashed vertical line), with the free space focus position taken as $z = 0$. Taking the difference between the focal position of the glass and spaceplate on glass cases shows that the spaceplate retracts the focus by 0.50 mm, equivalent to $\mathcal{R} = 43$. **b** For each of the above three cases, a lithographic defect comprised of gold on glass, shown in (**c**), was imaged at each of the focal positions found from **a**. For the spaceplate case, the sharp middle image indicates that this is indeed the image plane, demonstrating that the imaging system has been effectively shortened by 0.50 mm.

fabrication of free-standing thin-film optical filters has been demonstrated[23] and is now a viable, if cutting-edge, fabrication technology. However, there are still technical challenges associated with the process. Therefore, for this proof-of-principle demonstration, the thick glass simplifies fabrication and resists bending from the mechanical stress applied by the layers, which could add its own focus and change the magnification. However, the glass substrate complicates isolating the effect of the spaceplate since glass creates focal and lateral shifts opposite to those of a spaceplate. We find the shift due to the multilayer stack alone by measuring and subtracting the shift created by a bare glass substrate of identical thickness, as described in the caption of Fig. 2b.

We start by applying the above method to measure the focal shift caused by a spaceplate. In Fig. 3a, we present measurements of the location of the focus of a beam that has traveled through a lens, and then a spaceplate, similar to what is depicted in Fig. 1b. The 1546.85 nm wavelength beam has a Gaussian spatial profile and is focusing with an angular width of 1 degree. We find the focus's location along the propagation direction $z$ by recording the $x$–$y$ spatial intensity distribution at a sequence of $z$ positions using a camera. In Fig. 3a, we plot the spatial width $\omega$ along the $x$ direction found from these distributions. The minimum width, i.e., the beam waist, marks the position of the focus, which we find by fitting to a standard hyperbolic relation for a Gaussian beam[16]. We define the location of the focus with nothing in the beam's path as $z = 0$. When only the glass piece is in the pathway of the beam, it comes to a focus at $z = 1.07$ mm, which is further from the lens, as expected for a medium with $n > 1$. On the other hand, when the spaceplate on glass is present, the beam focuses at $z = 0.57$ mm, implying that the 11.51 $\mu$m-thick spaceplate shortens the focal distance by 0.50 mm.

While the advance of a focusing beam provides a quantitative demonstration of the spaceplate effect, the more relevant aspect is its

action on more structured fields, such as those present in imaging. For this purpose, a 100 x 100 $\mu m^2$ square section of a glass plate lithographically patterned with a gold structure was imaged. We chose a region containing a defect in the lithography as the imaging object. The gold-defect object is illuminated by the 1550 nm laser and imaged with a lens with a diameter of 25.4 mm, the focal distance of $f = 50$ mm, and a magnification of 1.53. We show camera images recorded at the three focal positions found for each scenario described above in Fig. 3b. The sharpest image for free space is found at $z = 0$, while glass only is $z = 1.07$ mm, and spaceplate on glass at $z = 0.57$ mm. This advancement was expected from the beam waist measurements, and the image has no apparent magnification. We thereby observe the demonstration of the compactification of an imaging system by the spaceplate.

We now discuss two peripheral effects observed in the image. First, the image contrast decreases when the spaceplate is inserted into the beam path. The spaceplate only transmits those $k$-vectors that lie in its working angle-range and, therefore, naturally acts as a low-pass filter (for more information, see Supplementary Method 4 and Supplementary Fig. 3). This is secondary to the spaceplate effect itself, and devices with higher NA will provide better image quality. Second, while a few features for the spaceplate on glass case may appear sharper at $z = 1.07$ mm than the focus at $z = 0.57$ mm, these are artifacts of defocus. Indeed, most features are sharper at the focus, $z = 0.57$ mm, and the shape of the artifacts does not match the object nor the free space image at its focus at $z = 0$ mm. These artifact sharp features often appear due to diffraction peaks in defocused images. Overall, the image with the spaceplate is better defined at $z = 0.57$ mm, as expected.

## A comparison of the performance of the different spaceplate designs

We test and compare four spaceplate devices, two based on the gradient-descent design (GD1 and GD2) and two based on the Fabry-Perot design (FPC1 and FPC2). In order to explore performance trade-offs, the GD1 and FPC1 devices were designed to have a larger $\theta_{max}$ than the GD2 and FPC2 devices. We investigate whether this higher angular range comes at the expense of bandwidth and compression ratio. We detail below how we characterize these performance parameters and discuss select results. A summary of the design and measured characteristics is given in Table 1. Note that, while FPC2 does not have the highest $\mathcal{R}$ of our devices, it was chosen for the focal shift measurements due to the large absolute size of its focal shift and overall performance stability of the device.

The most accurate way to determine the compression ratio $\mathcal{R}$ achieved by each device is to use the method shown in Fig. 2a and b. As depicted in Fig. 1a, the lateral beam shift[7] is given by

$$\Delta x = -(\mathcal{R} - 1)d\sin\theta. \tag{1}$$

Figure 4a shows the measurement of $\Delta x$ due to the spaceplate alone for all four devices. For each, the shift follows Eq. (1), increasing with angle up to a limiting angle, above which the shift gradually decreases to zero. This limiting angle defines the maximum angle of incidence $\theta_{max}$ of the spaceplate device, which is indicated by the shaded area for GD1 as an example. In the small-angle approximation, Eq. (1) for the lateral shift is a line, $\Delta x \approx -\theta(\mathcal{R} - 1)d$. We fit this line to the measured beam shift up to the limiting angle, thereby finding an experimental value $\mathcal{R}$ for each device. Both the small and large $\theta_{max}$ variants of either the GD or FPC designs exhibited the expected lateral shift of a spaceplate (see Supplementary Method 6 for fabrication details). That is, the shift increased linearly with angle and was in the opposite direction to the shift created by the glass. The lateral beam shift is proportional to the angular dispersion of the device transmission phase (See Supplementary Method 1). The measurements of the lateral shift as a function of angle effectively characterizing imaging

**Table 1 | Summary of the performance characteristics for spaceplate devices GD1, GD2, FPC1, and FPC2**

| Device | Thickness | Maximum Angle of Incidence $\theta_{max}$ | FWHM Bandwidth | Wavelength | Max Compression Ratio $\mathcal{R}$ | Theoretical Compression Ratio Limit $\mathcal{R}_t$ | Transmission |
|--------|-----------|------------------------|----------------|------------|-----------------|-----------------|--------------|
| | ($\mu$m) | | (nm) | (nm) | | | (%) |
| FPC2 | 11.51 | 1° | 0.143 ± 0.004 | 1532.9 | 96 ± 2 | 12,292 | 17 |
| FPC2 | 11.51 | 1° | 0.282 ± 0.006 | 1546.85 | 41.9 ± 0.6 | 6466 | 11 |
| FPC2 | 11.51 | 1° | 0.147 ± 0.005 | 1560.1 | 48.6 ± 1.4 | 12,860 | 42 |
| GD1 | 2.48 | 10° | 2.8 ± 0.3 | 1561.66 | 60 ± 4 | 658 | 6 |
| GD2 | 13.42 | 1° | 0.055 ± 0.007 | 1566.06 | 176 ± 14 | 33,562 | 73 |
| FPC1 | 12.57 | 3.5° | * | 1579.8 | 3.4 ± 0.3 | * | 5 |

For FPC2, we include values for each spectral response peak shown in Fig. 4**b**. The maximum compression factor and its uncertainty are, respectively, the mean and standard error over six trials. The wavelength and transmission noted here are recorded at the maximum compression ratio $\mathcal{R}$; the bandwidth is the peak's spectral width, while the maximum angle of incidence $\theta_{max}$ is from the device design. (*) Only the peak compression ratio was extracted for FPC1 before the device malfunctioned.

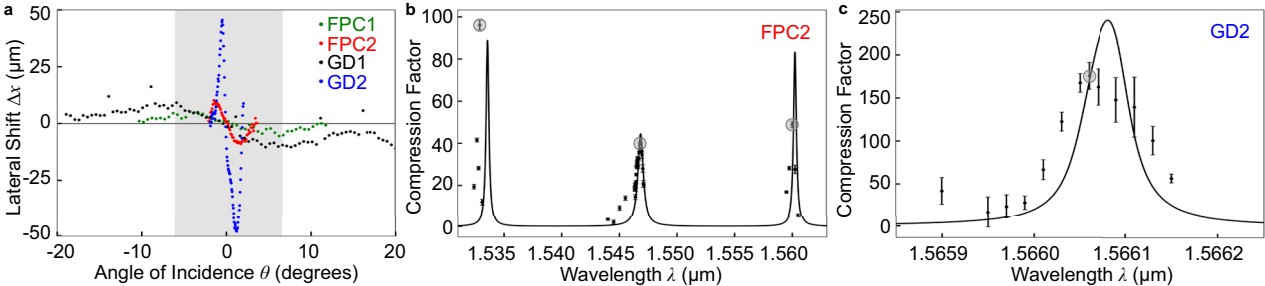

**Fig. 4 | Measured lateral shift and spectral dependence of the compression ratio for the spaceplate devices, GD1, GD2, FPC1, and FPC2. a** The lateral shift $\Delta x$ for four devices was fit to find $\mathcal{R}$(GD2) = 176 ± 14 (blue), $\mathcal{R}$(FPC2) = 41.9 ± 0.6 (red), $\mathcal{R}$(GD1) = 30 ± 3 (black), and $\mathcal{R}$(FPC1) = 3.4 ± 0.3 (green). The shaded region indicates the GD2's angular range from $-\theta_{max}$ to $+\theta_{max}$ as an example. **b, c** The compression ratio found from the fit (solid points) for devices FPC2 and GD2, respectively, for a given input optical wavelength $\lambda$, along with the compression ratio from the modeled phase response of the multilayer stack (solid black line). The peak measured compression ratio (grey circles) and other characteristic parameter values are given in Table 1. Error bars are one standard deviation, which is found from the covariance matrix of the nonlinear least-squares minimization fit.

performance in the same manner as ray-tracing, i.e., we are experimentally tracing a set of beams. In Fig. 3a, the angular dependence of the lateral shift is further tested directly on the collection of ray angles contained in a focusing beam, i.e., the focal shift $\Delta$.

The multilayer spaceplates feature frequency resonances over which both the transmission and the compression will vary (for further discussion, see Supplementary Method 5, and Supplementary Fig. 4). The GD designs have a single narrow resonance peak around the designed wavelength (Fig. 4c), while the FPC designs feature multiple peaks, each with different peak compression ratios (Fig. 4b). The experimental central wavelengths are offset compared to the target of 1550 nm, as we were targeting the demonstration of an optical spaceplate and not the wavelength accuracy in the fabrication. For future commercial spaceplates, it would be relatively simple to also target wavelength accuracy with a modern thin-film deposition system with active feedback. To determine the spectral bandwidth of the devices, we measure the transmission (see Supplementary Method 3 and Supplementary Fig. 4) and the compression ratio over the range of input wavelengths. The device bandwidth is then calculated as the full width at half maximum (FWHM) of the transmission curve.

Note that all devices are quoted for p-polarized light, although s-polarized light exhibits comparable performance in a limited angular range of $\theta_{device} \leq 10°$ (see Supplementary Method 7 and Supplementary Fig. 5 for details). As the discrepancy in the polarization response grows with NA, low-NA multilayer spaceplates may be more useful in preserving the shifted image's polarization. On the other hand, this polarization dependence in large NA spaceplates could be used as a polarization separator in applications such as reflection or glare reduction and in polarimetric imaging[24,25].

Table 1 compiles the performance metrics for all of the space-plates. The diversity of values in the table demonstrates the freedom of a designer to tailor a multilayer spaceplate for a specific application depending on the desired combination of $\theta_{max}$, bandwidth, and compression ratio. The designs based on gradient descent demonstrated the maximum performance for all three of the latter parameters: GD2 had the highest compression ratio ($\mathcal{R} = 176 ± 14$), whereas GD1 had both the highest bandwidth (2.8 ± 0.3 nm) and $\theta_{max}$ (10°). Trade-offs between these parameters were evident. For example, the highest compression ratio (GD2) was accompanied by a modest $\theta_{max}$ of 1° and the smallest bandwidth of 0.055 ± 0.007 nm of any of the designs. Additionally, the large $\theta_{max}$ of GD1 came with a smaller compression ratio, $\mathcal{R} = 60 ± 4$. A counter-example is that all three measured transmission peaks of device FPC2 demonstrated different compression ratios while retaining the same angular range $\theta_{max}$ of 1°. These results show that while there are often tradeoffs on the limits of the angular range, bandwidth, and compression ratio that need to be considered when targeting a specific spacelate application[12]. The theoretical limits on the compression ratio for the parameters that match each device are calculated using Eq. 7 of ref. 12 and are listed in Table 1. For more details, see Supplementary Method 2.

Although low NA and narrow bandwidth can be limiting for broadband imaging applications, approaches to the mitigation of these constraints have been explored in the literature. For instance, as high-index thin-film materials become available, thick devices with higher NA and moderate compression ratios will be possible[10]. While creating a broadband compressive device in the visible spectrum remains a challenge due to the necessary high refractive index contrast, an RGB implementation has been proposed[26], which can be used

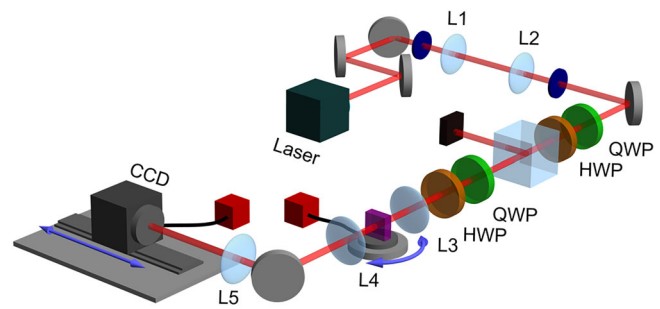

**Fig. 5 | Experimental setup to measure lateral and focal shifts.** A fiber-coupled tunable-wavelength laser creates a beam that is decreased in width by a telescope, lenses $L_1$ and $L_2$. Polarization optics (QWP = quarter-wave plate, HWP = half-wave-plate, PBS = polarizing beamsplitter) set the power and polarization of the beam incident on the spaceplate. The beam is focused by lens $L_3$ onto the spaceplate, which is on a rotation mount to vary the angle of incidence $\theta$. An object plane is imaged onto a camera by a 4-f lens system composed of two lenses, $L_4$ and $L_5$. The camera is translated so as to image different object planes along $z$ after the spaceplate.

in systems with narrow-band sources, like virtual reality (VR) headsets or laser cinema projectors.

## Discussion

In summary, we fabricated and characterized optical multilayer-stack spaceplates based on two design approaches: grouping layers into coupled resonators and optimizing layer thicknesses via gradient descent. We experimentally demonstrated that these devices can replace free-space propagation up to 176 times their thickness and, when integrated into imaging systems, reduce the distance between the imaging plane and optical components such as lenses. Since they exhibit sharp spectrally peaked responses, multilayer stack-based spaceplates are ideal for narrow-band imaging applications such as LIDAR[20] and retinal scanners. We demonstrated that a spaceplate can cancel the beam walk-off caused by a 260 times thicker glass plate, a feature that would be valuable in optical beam splitters[27] and advanced imaging systems[21,22]. In this way, our results demonstrate the potential of spaceplates for miniaturizing imaging systems and advancing flat optics.

Supplementary information accompanies this manuscript.

## Methods

The experimental setup is shown in Fig. 5. For all measurements, we use a continuous-wave 1.6 mW laser with a wavelength that can be tuned from 1525 nm to 1630 nm with a resolution of 0.1 pm. The beam exits a single-mode fiber and is collimated with an aspheric lens. After passing through a telescope composed of lenses $L_1$ and $L_2$, the beam has a beam width of 0.7 mm (All $L$ lenses are plano-convex). A quarter-wave plate (QWP), a half-wave plate (HWP), and a polarizing beam-splitter (PBS) are used to set the power of the beam at the spaceplate to 5 $\mu$W. Another QWP and HWP are used after the PBS to set the polarization input to the $p$-polarization with respect to the spaceplate surface.

### Measurement of the lateral beam shift

To measure the lateral beam shift $\Delta x$, we rotate the spaceplate about a vertical axis (i.e., $y$) using a motorized rotation stage, which varies the angle of incidence $\theta$. To explore the angular range of the spaceplate devices (Fig. 4a), the spaceplate was rotated in steps of 0.05° up to twice the design $\theta_{max}$ in both the positive and negative directions. To determine the compression ratio reported in Fig. 4b and c, the function was fit over the range of $-\theta_{max}$ to $+\theta_{max}$ to retrieve $\mathcal{R}$ for each input wavelength. (Throughout the paper, we quote $1/e^2$ intensity half-widths.)

Since the multilayer devices are of the order of ten microns thick, the size they create can be small. To measure the shift precisely, we reduce the beam size by focusing the beam with a third lens $L_3$ with focal length $f_3$ = 250 mm. The beam's width at the focus is $\omega_0$ = 190 $\mu$m (i.e., the beam waist). The beam passes first through the glass substrate and then the spaceplate multilayer device (nominally, this order should not matter). Note, even with the focusing, the beam's angular width 0. 2° is much smaller than any of the devices' designed $\theta_{max}$.

For each angle $\theta$ of the spaceplate, we determine the lateral beam position and, thus, shift $\Delta x$ by imaging the $x$–$y$ spatial intensity distribution of the beam. We set the object plane to be the focal plane of the beam, where our precision will be highest. Our imaging system consists of a 4-f telescope composed of lenses $L_4$ with focal length $f_4$ = 100 mm and $L_5$ with focal length $f_5$ = 150 mm. The measured magnification was M = 1.48. The imaged intensity distribution is measured with an Indium-Gallium-Arsenide (InGaAs) infrared charge-coupled device (CCD) camera (20 × 20 $\mu$m$^2$ pixels, with 320 x 256 pixels in the horizontal and vertical directions, Bobcat 320 Gig-E). We integrate the $x - y$ spatial intensity distribution over the $y$ direction and then fit to a Gaussian function in $x$ to find the location of the beam center. The difference between this location and the location at normal incidence is the, $\Delta x$.

### Measurement of the focal shift

To measure the focal shift, we image the $x - y$ spatial intensity distribution at different planes along $z$. To achieve the highest precision measurement of the focal position along $z$, we decrease the Rayleigh range of the beam by increasing its angular width to 1° by changing lens $L_3$ to have a focal length $f_3$ = 50 mm. Also, to increase precision, we change the imaging system magnification to M = 5 by switching $L_4$ and $L_5$ to focal lengths of $f$ = 50 mm and $f$ = 250 mm, respectively. The imaged object-plane is varied along $z$ in increments of 0.04 mm over a range of 2 mm by translating the camera with a motorized stage. At each $z$, we fit the $x$ marginal intensity distribution to a Gaussian to find the beam width $\omega$, as shown Fig. 3a.

### The spaceplate in an imaging system

We observe the action of the spaceplate devices in an imaging system. To do so, we form an image of a lithographic defect in gold using lens $L_3$ with a focal length $f_3$ = 50 mm and a diameter of 25.4 mm. The $z$ position of $L_3$ was such that a real image formed after spaceplate location with a magnification of M = 1.54 and an NA of 5.57°. We measure the $x - y$ spatial intensity distribution at different $z$-planes using the same parameters as in the focal measurement. When this $z$-plane coincides with the plane of the real image, the measured intensity distribution is that of the object, while at other planes it is blurry.

## Data availability

The authors declare that the data supporting the findings of this study and code used for data analysis are available within the paper, its supplementary information files, and in the repository: https://github.com/uOttawaQuantumPhotonics/Experimental-demonstration-of-high-compression-of-space-by-optical-spaceplates.

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

## Acknowledgements

The authors acknowledge support from the Canada First Research Excellence Fund award on Transformative Quantum Technologies (J.L., R.B.), Natural Sciences and Engineering Research Council of Canada (J.L., R.B.), Canada Research Chairs (J.L., R.B.), Banting Postdoctoral Fellowship of NSERC (O.R.), Vanier Canada Graduate Scholarships of NSERC (Y.M.), the National Natural Science Foundation of China (Grant No. W2533027) (R.H.), the Jiangsu Province Excellent Postdoctoral Fellowship Program (Grant no. 2025ZB498) (R.H.). We acknowledge Xiaoqin Gao and Francesco Monticone for scientific discussions, Jordan Pagé for the development of the gradient descent multilayer designs GD1 and GD2, Daniel Hutama for help with computerized experiment control, and Manuel Ferrer for feedback on the manuscript.

## Author contributions

O.R. conceived the experiment; R.H. designed and characterized the samples; B.S. and G.C. fabricated the samples; R.H. and O.R. simulated the results; R.H. and Y.M. conducted the experiment; R.H. and Y.M. analyzed the results and drafted the manuscript. R.C-C., J.L., and R.B. supervised the work. All authors discussed the results and contributed to the text of the manuscript.

## Competing interests

The authors declare no competing interests.
