## [Transparent Peer Review file · Nature Communications]

Experimental demonstration of high space compression by optical spaceplates

Corresponding Author: Dr Jeff Lundeen

Version 0:

Reviewer comments:

Reviewer #1

(Remarks to the Author)

This manuscript experimentally demonstrate multilayer-stack spaceplates designed to achieve a high compression ratio R in the optical wavelength region. A spaceplate is a nonlocal optical device that compresses the required free-space propagation length (d_{eff}) into a shorter physical thickness (d), with the compression ratio $R=d_{\text{eff}}/d$. The authors characterized fabricated devices designed using two methods: gradient descent (GD) optimization and a Fabry-Pérot Cavity (FPC)-based periodic structure. An exceptionally high compression ratio of $R=176$ is achieved. This work highlights the potential of these devices for compacting optical systems, particularly within narrow-band applications. The manuscript would benefit from further discussion on the following points to facilitate the audience's understanding.

- (1) Ref. 12 discusses trade-offs between the angular bandwidth, spectral bandwidth and compression ratio. However, as described in the manuscript, these trade-offs are not universally evident in your FPC devices. This is an interesting observation. Can you comment on the potential reasons? This might help others with future designs,
- (2) The compression ratio of $R = 176$ obtained experimentally is impressive. I am curious about how close it is to the fundamental bounds, for example, those established in Refs. 12, 13.
- (3) Including some results/comments on the polarization dependence of this device would be helpful, since it's a major concern for many applications.

Reviewer #2

(Remarks to the Author)

The manuscript "Experimental demonstration of high space compression by optical spaceplates" presents an experimental realization of compressing free-space propagation into a shorter distance by implementing multiplanar dielectric structures that reproduce a wavevector-dependent phase delay function in free space. This work can be considered an experimental follow-up to theoretical/numerical studies performed by the same authors (refs. 7 and 9). The authors present four spaceplates of two types: periodic and optimized. They demonstrate a large compression ratio of >170 , which means a free-space propagation distance can be shortened by that factor. The manuscript is well-organized, and the experimental validations, which involve measuring the transverse beam shift and axial focal spot shift, are convincing.

However, considering the experimental emphasis of this work, it could be improved by providing more information on the practical challenges of achieving such a large compression ratio and reducing the total size of optical systems. Especially, even with a compression ratio of >170 , the spaceplate only counteracts the effect of the 3 mm thick glass. Therefore, for spaceplates with a smaller compression ratio, the image plane is formed further away from the reference image plane than it would be in free space (as shown in Fig. 3), thus not shortening the total system length.

In this regard:

-Considering the trade-off relationship between angular range, bandwidth, and compression, can the compression ratio, R , be considered a fundamental quantity to characterize the device's performance? Given the small operable angle with the demonstrated bandwidth, what is the theoretically achievable R ?

-I am also concerned with the emphasis on a large R without clearly stating its small angular range, which would impede its practical usability.

-I suspect the use of 3 mm thick glass is due to unavoidable fabrication challenges. If that is the case, it would be useful to provide the key parameters that hinder the use of thinner glass layers. Specifically, information on the tolerance for height, curvature, and refractive index in the fabrication process would be helpful. Otherwise, I suggest including results with thinner glass, which might result in a more straightforward demonstration of the reduction of the imaging system's size, as the authors claim.

-Relevant to the previous question, does the requirement for fabrication tolerance become more stringent when achieving a larger angular range? Practically, an R as small as 10 but with a larger angular range would be much more useful for imaging systems.

-In this regard, larger-NA imaging experiments with GD1 design would be more compelling in showing the reduction in the system's total size.

-It would also be better to provide the angular phase and amplitude responses of GD1 and GD2 (which would be particularly important to judge if the devices are operating as spaceplates) and their optimized index profiles along the z-axis.

-Looking at the imaging results in Fig. 3b, the "spaceplate+glass" image does not appear to be sharply focused at $z \approx 0.5$ mm. Looking closely, some features appear sharper at $z \approx 1$ mm (the same plane as with a bare glass). Why is that the case? Is it related to the non-ideal phase response of the FPC2 design?

Reviewer #3

(Remarks to the Author)

Ryan Hogan et al. present the experimental demonstration of optical path compression with specific spaceplates. This manuscript reports the use of spaceplates in the near-IR to shrink the optical path of conventional refractive lenses. Nowadays, reducing the optical path is an important issue for decreasing the form factor of optical devices such as VR/AR and LiDAR systems, including wearable devices. The authors demonstrate the possibility of using spaceplates at NIR wavelengths for this purpose. However, I believe the overall quality of the work does not fully meet the threshold for Nature Communications. Therefore, I cannot recommend publication in this journal and instead recommend submission to another journal. Below are some specific comments related to this work.

Major comments

1. The authors claim that the device can be freestanding but was bonded to glass for a proof-of-principle demonstration. This has not been substantiated. As a result, it remains unclear whether the approach is practically applicable to reduce optical path.

2. When the focal length is changed using the spaceplate, the beam width appears to increase, and the image quality is much worse than in free space. What is the cause of this degradation? Given this outcome, it is not clear that reducing the optical path provides meaningful advantages at the system level.

3. Even considering the designed spaceplate alone, there are issues. Perhaps because the device relies on resonances, the transmission is too low for practical use. More importantly, the fabricated spaceplate is significantly detuned from the target wavelength. As the authors' own data indicate, spaceplates typically have very narrow bandwidth; failing to realize the device at the target wavelength is a substantial problem.

Minor comments

In Fig. 2a, the refraction angle and behavior of light inside the glass appear to be physically implausible. This could confuse readers.

Reviewer #4

(Remarks to the Author)

Version 1:

Reviewer comments:

Reviewer #1

(Remarks to the Author)

The authors have addressed all comments, and the manuscript has been revised appropriately. The contribution and broad context of the work are now clearer. I have no further comments.

Reviewer #2

(Remarks to the Author)

The revised manuscript now acknowledges general limitations of the demonstrated experiment, including the requirement of a glass layer and comparisons to theoretical compression ratios. However, my reservations regarding practicality remain, as the authors have not provided plausible additional experimental data to address this critical point.

Regarding the imaging experiment with GD1: even with smaller focal shifts, the high NA should make the distinction between the 'space plate + glass' and 'glass-only' cases clearer, as the image would defocus much more rapidly from the ideal plane. I am still not fully convinced that Figure 3 demonstrates that the 'space plate + glass' achieves imaging quality comparable to the 'glass' control with a shortened track length.

I think that the definition and purpose of a space plate lie not in creating an exotic beam walk-off effect, but in producing an angular dispersion effect equivalent to a much longer free-space path.

While the beam walk-off experiments in Figure 4 are somewhat compelling in the sense that the space plate mechanism is functioning, they do not explicitly show that the space plate produces the specific angular dispersion effect required to shorten the imaging device in a practical sense.

Reviewer #3

(Remarks to the Author)

The authors have successfully resolved all concerns previously raised. I recommend the manuscript for publication, particularly recognizing the first-ever usable spaceplate.

Reviewer #4

(Remarks to the Author)

Version 2:

Reviewer comments:

Reviewer #2

(Remarks to the Author)

While the practical applicability of this work would benefit from further investigation—such as a detailed assessment of experimental beam walk-off or an imaging demonstration using space plates at a higher angular range—the authors' current revisions are adequate, given that the experimental setup has been disassembled. I therefore recommend the manuscript for publication, highlighting its significant contribution as the first experimental demonstration of space plates.

Response Letter (Manuscript #: NCOMMS-25-77097)

We would like to thank the editor, and reviewers for their care in handling our manuscript and their valuable and constructive feedback, and thoughtful reviews. The following is a letter regarding the resubmission of our manuscript, “*Experimental demonstration of high space compression by optical spaceplates*” (Manuscript #: NCOMMS-25-77097) for consideration of publication in Nature Communications.

All three referees appeared to recognize the potential impact of the topic of our manuscript, the reduction of optical path, as highlighted by Referee 3, who wrote that it “is an important issue for decreasing the form factor of optical devices”. Referee 1 highlighted that “the compression ratio of $R = 176$ obtained experimentally is impressive” and noted that the manuscript “highlights the potential of these devices [spaceplates] for compacting optical systems”. Referee 2 mentioned that the manuscript is “well-organized, and the experimental validations, which involve measuring the transverse beam shift and axial focal spot shift, are convincing”. Nonetheless, all three referees had significant reservations about the paper, particularly regarding the theoretical limits of our designs, image quality and the challenges related to the fabrication of the devices on different substrates. Consequently, we have made important revisions to the manuscript accordingly. We briefly address the primary issues here, while in the appendix, we give detailed responses to each specific comment and suggestion of each of the referees.

We begin by addressing practical issues arising from the limited angular and spectral bandwidth. With higher index thin-film coatings becoming available (e.g. titanium dioxide), simple repeating designs will allow for an NA that is practical for certain imaging systems (Chen, A. & Monticone, F. ACS Photonics, 8, 5, 1439–1447 (2021)). Regarding bandwidth, narrow bandwidth spaceplates could be useful in laser-illuminating imaging systems, which appear in a wide variety of contexts, such as LIDAR and retinal scanners. A recent RGB narrowband design (Reference?) demonstrated that spaceplates could be compatible with displays or projectors that use narrow-band RGB illumination (e.g. laser-illuminated VR goggles and SONY’s cinema laser projection system). We have added a section discussing these approaches to our Results section.

Another concern was the observed image deterioration when using a spaceplate. This observation is secondary to the spaceplate effect and is solely a loss in resolution due to the transmission NA of the device. As the spaceplate only transmits a range of accepted angles, it naturally acts as a low-pass filter, which reduces the contrast of the image. As a result, tackling the low NA of the system with the above designs will address any image quality concern as well. A discussion on this matter has been added to the Results section of the revised manuscript.

The referees also requested information about the polarization response of the devices. We have simulated such polarization responses for our devices and added a corresponding discussion in the Results section of the manuscript, as well as in Supplementary Method 7. While there is a minimal difference in polarisation response for the low-NA devices, we note a 40% change in the achieved compression factor for the higher-NA devices, which could be of interest for glare reduction or polarimetric imaging applications.

A comparison between observed performance of our fabricated spaceplates to previously a couple of published theoretical limits was requested. As a result, we have calculated the theoretical limits of the compression ratio for our devices and provided these results in Table I. In all cases, our measured

compression ratios are orders of magnitude lower than the theoretical limits suggesting that while our results improve upon previous records in literature, further design improvements are possible.

Comments regarding the need for the 3-mm thick substrate for our devices were also raised by the referees. Here, using a relatively thick substrate is a standard approach in thin-film stacks when characterizing innovative new designs, since it simplifies fabrication and reduces the effect of stress due to the thin-film stack. Nominally, stress or surface curvature could act as a lens that would complicate isolating the effect of the spaceplate, so meticulous care was taken to minimize these potential confounding factors. To further reduce the potential effect of stress, stress-compensating AR coatings were applied to all surfaces, excluding the surfaces comprising the spaceplates. Etalon-quality substrates were also used ($< \lambda/100$ flatness and parallelity). For comparison, we calculate that a curvature of 12λ would have been required to mimic the focal shifts we observed (this is a worst-case scenario, i.e., if the spaceplate were adjacent to the lens rather than at the focus). Future work will explore the use of thinner substrates and substrate-less thin-films. However, with our first-ever fabrication of a thin-film spaceplate presented here, utilizing the above ensured a reliable fabrication procedure.

The referees also asked for details about fabrication tolerances. Modern thin-film commercial deposition systems use an active feedback loop that monitors spectral transmission while depositing. This results in a layer thickness precision that far exceeds the minimum $\lambda/4$ at 1550 nm thickness required for our devices. We have added these discussions to the Introduction section of the manuscript and Supplementary Method 1.

We would like to emphasize that the importance of this work lies in fabricating a usable spaceplate for the first time, one that replaces space in vacuum in the optical region. Although theoretical designs have been proposed, practical implementations have not been demonstrated. We are presenting a device based on a well-developed fabrication platform compatible with modern optics, whose characteristics can be modified using standard techniques in thin-film optical design. The existence of these devices not only promises a reduction of the imaging systems, but also opens the door to the fabrication of other non-local optical elements, bringing the fully universal imaging transformation closer.

We hope that the above summary of the revisions, as well as our detailed response to each of the referees' comments below adequately addresses all concerns. Enclosed with this letter, we provide an annotated version of the revised manuscript and supplementary with changes highlighted in red. We believe that the comments and suggestions have improved the quality of our manuscript. We ask the reviewers to kindly reconsider our revised manuscript. Since imaging plays an extremely wide and far-reaching role in science, we expect our work will attract a broad readership and we hope that our manuscript is now suitable for publication in the widely-read journal of Nature Communications.

Yours Sincerely,

Ryan Hogan, Yaryna Mamchur, R. Margoth Córdova-Castro, Graham Carlow, Brian T. Sullivan, Orad Reshef, Robert W. Boyd and Jeff S. Lundeen

Response to Reviewer's Comments

The following contains detailed responses to each reviewer. Reviewer comments are presented in black, italic font, our response written in blue, and all revisions in the revised manuscript are highlighted in red in the attached file.

Reviewer 1:

Ref. 12 discusses trade-offs between the angular bandwidth, spectral bandwidth and compression ratio. However, as described in the manuscript, these trade-offs are not universally evident in your FPC devices. This is an interesting observation. Can you comment on the potential reasons? This might help others with future designs,

We thank the reviewer for the thoughtful question. The mentioned trade-offs are on the limits of the angular bandwidth, spectral bandwidth and compression ratio and not on the particular values exhibited by our devices. **The phrasing of the text has been clarified in the Results section of the revised manuscript (2nd last paragraph of Results section, top right, Page 4).**

The compression ratio of $R = 176$ obtained experimentally is impressive. I am curious about how close it is to the fundamental bounds, for example, those established in Refs. 12, 13.

We thank the reviewer for insightful comment. The fundamental limit for a thin-film device at 1550 nm made with silica and silicon and with our current device's bandwidth is $R = 33562$. The calculation was performed using formula 7 of Ref. 12 (Shastri et al., 2022). In all cases, our measured compression ratio, albeit larger than previously recorded in literature, is orders of magnitude lower than its theoretical limits, suggesting that there is ample room for improvements to the design. The fundamental limits from Ref. 13 (Miller, 2023) are not applicable to our devices, as Miller mentions in their work that "Published work on space plates has mostly not considered the number of pixels explicitly, making comparisons of our limit with some of this published work difficult".

The theoretical limits for all the devices have been added to Table I of the revised manuscript, and the calculation details are listed in Supplementary Method 2 of the revised supplementary materials (Sixth paragraph of Page 2, top right, Supplementary Information).

Including some results/comments on the polarization dependence of this device would be helpful, since it's a major concern for many applications.

We thank the reviewer for the suggestion. We have calculated the polarization response for all of our devices for both s- and p-polarized incident light. The maximum compression ratio discrepancy for the low-NA devices (GD2, FPC1) reaches 10 %, while higher NA devices (GD1, FPC2) reach up to 40%. **The results have been added as Supplementary Figure 5 of the revised supplementary materials (Page 4, bottom right, Supplementary Information). A discussion has also been added to the Results section of the revised manuscript (3rd last paragraph of Results section, bottom left, page 4).**

Reviewer 2:

Considering the trade-off relationship between angular range, bandwidth, and compression, can the compression ratio, R , be considered a fundamental quantity to characterize the device's performance?

We thank the reviewer for the thoughtful question. We believe R is indeed the spaceplate figure of merit parameter. It is uniquely used by spaceplates, specifically in the condition of $R > 1$. R can be calculated directly from the slope of the transverse walk-off, which depends on the wavelength and angle. R as a figure of merit can be compared to the Q-factor of the resonator. While the resonators have multiple tunable parameters. Depending on the application, a high Q-factor is useful for a variety of cases, and is indeed also inherently dependent on bandwidth and numerical aperture. As such, we expect that higher R will most likely still be the goal for many cases of its subsequent application, while spaceplate bandwidth and NA can still be tuned to adhere to specific applications. That being said, we agree that R should be stated alongside the bandwidth and the NA. **The values for R , NA, and bandwidth are stated in Table 1 of the revised manuscript (Top of page 5).**

Given the small operable angle with the demonstrated bandwidth, what is the theoretically achievable R ?

We thank the reviewer for their question. According to Eq. (7) of Shastri et al. (2022), the fundamental limit for a thin-film device made with silica and silicon designed for 1550 nm, with our corresponding device bandwidth, is $R = 33562$. The measured and theoretical vary in orders of magnitude, and thus the present designs can be further optimized to reach such large theoretical compression ratios. In any case, our current measured compression ratios surpass previously recorded, and support usage of spaceplates in the optical region. **The theoretical limits for all devices have been added to Table I of the revised manuscript (Top of page 5), and calculation details are listed in Supplementary Method 2 (Sixth paragraph of Page 2, top right, Supplementary Information) of the revised supplementary materials.**

I am also concerned with the emphasis on a large R without clearly stating its small angular range, which would impede its practical usability.

We thank the reviewer for their insightful concerns. The angular ranges for all devices are discussed in the results section of the revised manuscript. We have included a subsection entitled "A comparison of the performance of the different spaceplate designs", as well as included additional details in Table I. **A clarifying comment has been added to the Introduction section (2nd paragraph on page 2, top right), as well as a subsection describing a performance comparison to the results section of the revised manuscript (Bottom right on page 3 to the End of the Results section on page 4).**

I suspect the use of 3 mm thick glass is due to unavoidable fabrication challenges. If that is the case, it would be useful to provide the key parameters that hinder the use of thinner glass layers. Specifically, information on the tolerance for height, curvature, and refractive index in the fabrication process would be helpful. Otherwise, I suggest including results with thinner glass, which might result in a more straightforward demonstration of the reduction of the imaging system's size, as the authors claim.

We thank the reviewer for pointing this out. Standard approaches often use relatively thick substrates when creating thin-film stacks, especially when characterizing innovative new designs since it simplifies fabrication and reduces the effect of stress resulting from the thin-film stack. Stress or surface curvature could act as a lens that would complicate isolating the effect of the spaceplate, so we took great care to minimize these potential confounding factors. Stress-compensating AR coatings were applied to all surfaces, except the spaceplate itself, to further reduce the potential effects of stress. Etalon-quality substrates were also used ($< \lambda/100$ flatness and parallelity). For comparison, we calculate that a curvature of 12λ would have been required to mimic the focal shifts we observe (this is a worst-case scenario, i.e., if the spaceplate were adjacent to the lens rather than at the focus). Overall, the use of a 3 mm substrate is not necessary generally speaking, but provides a reliable fabrication procedure basis from which could test our spaceplates and demonstrate the first-ever fabrication of a thin-film spaceplate. Future work will explore the use of thinner substrates and substrate-less thin-films, referencing Roth et al. (2005) as support to these potential research pathways. **We have added these discussions to the Results section in the revised manuscript (1st paragraph of Results section, page 2, bottom right) and added the technical details to the Supplementary Method 1 of the revised supplementary materials.**

Relevant to the previous question, does the requirement for fabrication tolerance become more stringent when achieving a larger angular range? Practically, an R as small as 10 but with a larger angular range would be much more useful for imaging systems.

We thank the reviewer for their concerns regarding fabrication tolerances. The fabrication of a device with a larger angular range necessitates more layers and, thus, thicker thin film designs to achieve the required phase properties. This is well within the capabilities of current commercial fabrication, which can deposit up to 2000 layers accurately using realtime feedback. Achieving higher bandwidth and angular width with thicker devices is a design path commonly used in thin-film optical filters and other theoretical designs of the spaceplate (Chen and Monticone, 2021). **A discussion on broadening the angular range of the devices has been added to the Results section of the revised manuscript (4th paragraph of Results section, bottom left, page 3).**

In this regard, larger-NA imaging experiments with GD1 design would be more compelling in showing the reduction in the system's total size.

We thank the reviewer for their comment. Indeed, GD1 has a higher NA, however it is also much thinner than other demonstrated devices. With a compression ratio of 60 and the thickness of $3.55 \mu\text{m}$ it would create the focal shift of $213 \mu\text{m}$, while FPC2 at 1546.85 nm provides us with the compression ratio of 41.6 with the device thickness of $12.04 \mu\text{m}$, resulting in the $504 \mu\text{m}$ focal shift. We have chosen the device with a higher absolute focal shift to be highlighted in the manuscript. **A clarifying comment has been added to the Results section of the revised manuscript (last paragraph on page 3, bottom right).**

It would also be better to provide the angular phase and amplitude responses of GD1 and GD2 (which would be particularly important to judge if the devices are operating as spaceplates) and their optimized index profiles along the z-axis.

We thank the reviewer for their insight. For the angular phase response, the transverse walk-off negative slope in Fig. 4a already contains phase information indirectly through the relationship from

Supplementary Method 1 $w = \frac{-1}{k \cos\theta} \left(\frac{\partial\phi_t}{\partial\theta} \right)_{\theta_0}$. As Fig. 4 is already informationally dense, an example

of this relationship is provided in the Supplementary Method 1. Regarding the amplitudes, index profiles of GD1 and GD2 have been included in the revised manuscript for improved clarity and transparency of our devices. **The corresponding index profiles have been added to the Supplementary Table II in the revised supplementary materials (Page 5, Supplementary Information), and the relationship is demonstrated in Fig. S1 of the revised supplementary materials (Page 1, bottom right, Supplementary Information), and a discussion has been added to the Results section of the revised manuscript (1st paragraph of page 4, top left).**

Looking at the imaging results in Fig. 3b, the “spaceplate+glass” image does not appear to be sharply focused at $z \approx 0.5$ mm. Looking closely, some features appear sharper at $z \approx 1$ mm (the same plane as with a bare glass). Why is that the case? Is it related to the non-ideal phase response of the FPC2 design?

While a few features for the “spaceplate on glass” case may appear sharper at $z = 1$ mm than the focus at $z = 0.5$ mm, these are artifacts of defocus. Indeed, most features are sharper at the focus, $z = 0.5$ mm and the shape of the artifacts do not match the object nor the free space image at its focus at $z = 0$ mm. We attribute these artifact sharp features to diffraction peaks, which often appear in defocused images. **A discussion clarifying the above has been added to the Results section of the revised manuscript (4th paragraph of page 3, bottom right).**

Reviewer 3:

The authors claim that the device can be freestanding but was bonded to glass for a proof-of-principle demonstration. This has not been substantiated. As a result, it remains unclear whether the approach is practically applicable to reduce optical path.

We thank the reviewer for their insightful comment. Regarding the free-standing spaceplate, the ability to fabricate free-standing thin film optical filters has been accomplished previously (Roth et al., 2005). There are still technical challenges associated with free-standing filters, however, it should be possible. Indeed, we have fabricated some free-standing prototype spaceplates for future testing. Thinner glass is also certainly possible but, for a first demonstration, we found that using thicker glass would remove any complicating effects, leading to more reliability in our measurements. In particular, it reduces the focusing effect of curvature in the substrate, which could lead to misinterpretation of results.

In Fig. 2b specifically, we demonstrate the cancellation of the walkoff effect of light propagation in a glass substrate. Glass takes up much of the space in imaging systems and often, is necessary as a protective window, e.g. as a sensor cover or vacuum system window. Thus, we show a clear advantage in some imaging situations, demonstrated explicitly. Implicitly, the optics of glass windows is well understood, so we argue that one does not need to substantiate that a spaceplate will work on a thinner substrate. As usual, as in any development process, small technical issues may arise. We chose to avoid these for a first demonstration.

A discussion has been added to the Results section (1st paragraph of Results section, page 2, bottom right) and Supplementary Method 1 (Page 1, Supplementary Information) of the revised manuscript.

When the focal length is changed using the spaceplate, the beam width appears to increase, and the image quality is much worse than in free space. What is the cause of this degradation?

We thank the reviewer for their concerns. A spaceplate only transmits k-vectors that lie in its working angle range and therefore, naturally acts as a low-pass filter. Even though the spaceplate k-width is matched to the beam k-width, there will still be narrowing of the range of k-vectors in the beam and a corresponding broadening of the beam width. We calculate the ratio of 0.96 between the beam waist in free space and after the spaceplate, which agrees with our experimental observation of 0.95. Low-pass filters also decrease contrast, which is the main difference from the free-space image. **Supplementary method 4 has been added to the revised supplementary materials (Page 3, right column, Supplementary Information) describing the details of the broadening effect and its corresponding calculations.**

Given this outcome, it is not clear that reducing the optical path provides meaningful advantages at the system level.

We thank the reviewer for the comment. We would like to point out that this reduction in quality is secondary to the spaceplate effect itself. Devices with higher NA will provide better image quality that could benefit systems such as retinal scanners, LIDAR, and VR headsets.

Even considering the designed spaceplate alone, there are issues. Perhaps because the device relies on resonances, the transmission is too low for practical use.

We thank the reviewer for their concerns. The transmission on resonance can be high, as in devices GD2, FPC2 (near 1560 nm). Theoretical designs (Chen & Monticone, 2021) show that highly transmissive devices can be achieved with high NA and/or bandwidth for useful values of R, much like in the design of standard thin-film optical bandpass filters. On the other hand, there are also valuable narrowband applications, including VR headsets or laser projection systems.

More importantly, the fabricated spaceplate is significantly detuned from the target wavelength. As the authors' own data indicate, spaceplates typically have very narrow bandwidth; failing to realize the device at the target wavelength is a substantial problem.

We thank the reviewer for their comment. We acknowledge the discrepancy between the target and achieved central wavelength, however, we have chosen to proceed with our experimental demonstrations for two main reasons. Firstly, we believe that the offset does not hinder the ability of our devices to demonstrate the spaceplate effect, as all our measurements were processed according to the measured central wavelength. In addition, it is known that thin film deposition systems with appropriate thickness control monitoring are capable of precise wavelength targeting, as multiple commercial multilayer optical filters are available on the market. **To address this concern, we have added a discussion in the Results section of the revised manuscript (2nd paragraph, page 4, middle left).**

In Fig. 2a, the refraction angle and behavior of light inside the glass appear to be physically implausible. This could confuse readers.

We thank the referee for noticing this mistake. In our attempt to exaggerate the distances on the schematic to exemplify the contrast of each case, we exaggerated the angles too much. **As a result, we have updated Fig. 2a to be physically correct in the revised manuscript (Page 2, top left).**

Response Letter (Manuscript #: NCOMMS-25-77097A)

We would like to thank the editor and reviewers for their extended care in handling our revised manuscript. We believe the inclusions have improved the overall quality of the manuscript. The following is a letter regarding the resubmission of our manuscript, “*Experimental demonstration of high space compression by optical spaceplates*” (Manuscript #: NCOMMS-25-77097A) for consideration of publication in Nature Communications.

We are pleased to hear that Reviewers 1, and 3 are satisfied with our previous revisions. We also would like to thank Reviewer 4 for the added assessment of our manuscript. Moreover, we appreciate the concerns of Reviewer 2 regarding practicality and the overall designs, and insights about additional experimental details. Below we address Reviewer 2’s concerns to the best of our current ability. Since our experimental apparatus has been disassembled and students are no longer here to perform the experiment requested by Reviewer 2, taking further data to explore the spaceplate’s performance will need to be the topic of a future project in our lab. We stress that the presented data in our current version of the manuscript is sufficient to prove the spaceplate performs as intended. Namely, we provide three distinct types of experimental data, the beam walk-off, the image plane advance, *and* the advance of the focal plane. The latter is not mentioned by Reviewer 2. These three types of data, indeed the entire principle behind the spaceplate, are given by angular dispersion of the phase response. That is they characterize exactly the same response but in different ways.

The following contains detailed responses. Reviewer comments are presented in black, italic font, our response written in blue.

Reviewer 2:

Regarding the imaging experiment with GD1: even with smaller focal shifts, the high NA should make the distinction between the 'space plate + glass' and 'glass-only' cases clearer, as the image would defocus much more rapidly from the ideal plane. I am still not fully convinced that Figure 3 demonstrates that the 'space plate + glass' achieves imaging quality comparable to the 'glass' control with a shortened track length.

We agree that for high-NA systems, deviations from the ideal focal plane would be more pronounced. However, we also had to consider the range of propagation distance that we could scan across with our motorized stage and IR camera. With these experimental constraints FPC2 was the spaceplate with the largest focal shift that we could observe with our setup.

As a proof-of-principle, we believe the imaging experiment demonstrates the desired spaceplate effect, the retraction of the image plane and the formation of a distinct image. We have performed additional numerical analysis of the angular dependence of the focal shift for GD1, GD2 and FPC2 devices and added the findings to Supplementary Information in Supplementary Method 8. For FPC2 the focal shift is uniform over approximately the full device angular range, supporting that the image quality is primarily due to the spatial low-pass filter effect we discussed in previous revision.

I think that the definition and purpose of a space plate lie not in creating an exotic beam walk-off effect, but in producing an angular dispersion effect equivalent to a much longer free-space path.

While the beam walk-off experiments in Figure 4 are somewhat compelling in the sense that the space plate mechanism is functioning, they do not explicitly show that the space plate produces the specific angular dispersion effect required to shorten the imaging device in a practical sense.

The reviewer has reservations about relying on beam walk-off data and also about the quality of the image plane data. However, the manuscript provides a third type of data that corroborates the conclusions we draw from those two datasets, namely, the measured longitudinal shift in a focusing beam (Fig. 3a). All three types of data quantitatively agree. Taken together, the three characterization methods provide convincing evidence of the primary effect of the spaceplate. That is, we observe the first optical replacement of free-space propagation in vacuum by a thin device.

We emphasize that the angular dispersion and the beam walk-off are one in the same. Namely, the angular derivative of the transmission phase gives the walk-off, $w = \frac{-1}{k \cos \theta} \left(\frac{\partial \phi}{\partial \theta} \right)$. Imaging is independent of a global phase, so this gradient is the underlying cause of the spaceplate effect and completely characterizes it [more details can be seen in Supplementary Materials of *Nat. Commun.* **12**(1), 3512 (2021)]. By experimentally showing that the beam walk-off is linearly dependent on angle (Fig. 2b and 4a), we are characterizing imaging performance in the same manner as ray-tracing, i.e., we are experimentally tracing a beam. This angular dependence of w is further tested directly on the collection of ray angles contained in a focusing beam, i.e. the focal shift $\Delta \propto \frac{\partial w}{\partial \theta}$. We experimentally observe a transverse walk-off with a negative linear slope and the retraction of the focal plane of the lens. These two types of data are much more amenable to quantification than imaging, they quantitative agree with each other, they fundamentally are the angular dispersion of the

phase that is behind the spaceplate effect, and they support our observation that imaging plane is retracted toward the lens by the expected amount.

We thank the reviewer for drawing our attention to the fact that this connection was not clear enough in the manuscript. We have added the following sentences to explicitly address this (lines 237-243):

The measurements of the lateral shift as function of angle are effectively characterizing imaging performance in the same manner as ray-tracing, i.e., we are experimentally tracing a set of beams. In Fig. 3a, the angular dependence of the lateral shift is further tested directly on the collection of ray angles contained in a focusing beam, i.e., the focal shift Δ .